# A Cooperative Management App for Parents with Myopic Children Wearing Orthokeratology Lenses: Mixed Methods Pilot Study

**DOI:** 10.3390/ijerph181910316

**Published:** 2021-09-30

**Authors:** Chi-Chin Sun, Gen-Yih Liao, Li-Ling Liao, Li-Chun Chang

**Affiliations:** 1Department of Ophthalmology, Chang Gung Memorial Hospital, Keelung 204, Taiwan; arvinsun@cgmh.org.tw; 2College of Medicine, Chang Gung University, Taoyuan City 333, Taiwan; 3Department of Nursing, Chang Gung Memorial Hospital, Taoyuan City 333, Taiwan; gyliao@mail.cgu.edu.tw; 4Department of Information Management, Chang Gung University, Taoyuan City 333, Taiwan; 5Department of Health Management, I-Shou University, Kaohsiung City 824, Taiwan; hililin@isu.edu.tw; 6School of Nursing, Chang Gung University of Science and Technology, Taoyuan City 333, Taiwan

**Keywords:** orthokeratology lens (OK), compliance, myopic children, parents, mobile application

## Abstract

Orthokeratology (OK) lens wear is an effective modality to inhibit axial elongation in myopic children. Willingness for commitment from both parents and children contributes to the success of OK treatment. We aimed to develop and assess the usability of a mobile application on OK lens wear by quantitatively and qualitatively evaluating parents with myopic children and eye care professionals (ECPs). Moreover, the preliminary outcome was also evaluated in this study. The app was developed and tested using a co-design approach involving key stakeholders. Two prototype tests were conducted during the feasibility and utility assessment. The app features include self-reported compliance documentation, analytics, and personalized and generalized messages for compliance behaviors of OK lenses. After the trial period, the full usage of app functions ranged from 40% to 60% among the enrolled parents. After app implementation, the compliance with follow-up visits substantially improved. Qualitative data show that the high-satisfaction app functions reported by parents were the app’s reminder and axial length recording, although it was recommended that the number of compliance questions should be reduced to minimize the survey completion time. Additionally, who should complete the recording of the axial length data as well as the management and reminder for the follow-up visit remained controversial. This is the first app developed to improve parents of myopic children’s compliance with OK lens wear and to assist ECPs and parents in collaboratively monitoring and managing the use and care of OK lenses among myopic children. This study highlights the importance of interdisciplinary collaboration in the design, development, and validation of such an app.

## 1. Introduction

Myopia is an underappreciated but profound public health problem that affects almost 30% of the world’s population. Of the ever-growing global myopic population, 312 million people are under the age of 19 years [1]. Excessive axial elongation is associated with myopia, and this leads to structural changes in the posterior segment of the eye (including posterior staphyloma, myopic maculopathy, and high myopia-associated optic neuropathy) that may culminate in loss of best-corrected visual acuity in adulthood [2]. In addition, aging and the high utilization of touchscreen technology might increase the risk of eye diseases [3]. Therefore, countries with a high myopia prevalence would face significant medical costs and compromised population productivity [3,4]. A systemic review indicated that a young age of onset and a fast progression rate during childhood are important risk factors of high myopia [4], prompting the World Health Organization (WHO) to call for attention on controlling myopia among school children [1].

The World Report on Vision [1] has suggested that it is important to continuously monitor and manage myopia in children and adolescents, and that aggressive strategies are needed to slow the progression of high myopia, thereby minimizing the damage caused by the disease [5]. Orthokeratology (OK) is the process of deliberately reshaping the anterior cornea by utilizing specialty contact lenses to temporarily and reversibly reduce refractive error after lens removal [6]. A systemic review suggested that the use of OK lenses is the only method that can simultaneously decrease the progression of myopia and provide clear, unaided vision during the day [7]. In addition, OK lenses are more effective at controlling axial length growth in children aged 8–12 years than in adolescents and adults [8]. Studies have revealed that not only are OK lenses 40–60% more effective than other optical correction methods in reducing axial length growth [8] but they are also being increasingly accepted by these children and their parents, especially in East Asian countries [9]. Therefore, the popularity of OK lenses with ophthalmologists, patients, and parents has been steadily increasing [10].

The aftercare schedule for OK lens patients is more intensive than that for conventional lens wear patients, especially during the first year [11], and it is essential that patients, particularly children, return for the scheduled aftercare to minimize complications and to enhance OK lens compliance [6]. Moreover, treatment success and the decrease in the chances of complications due to non-compliance are management indicators for OK lenses [12,13]. Regular monitoring of OK lenses during follow-up visits could aid in monitoring the growth of the axial length as well as reducing the risk of infection [14]. However, the risk of non-compliance among young children in wearing OK lenses is considerably higher than that of other age groups [15], which can affect the achievement of visual acuity among children [16]. Furthermore, non-compliance with the lens wear schedule may have a detrimental effect on the control of myopia progression. Indeed, a higher compliance with the lens wear schedule has been reported to result in greater myopia control efficacy [17]. Poor compliance with lens hygiene and inadequate follow-up were risk factors for infection [18]. Parents are primary gatekeepers of their children’s health behaviors [19]; however, reaching and engaging parents in compliance behaviors can be challenging.

The use of mobile apps helps increase treatment adherence, and such apps are an appropriate method for managing medication at home [20]. Previous studies revealed the effectiveness of mobile apps in school children’s vision health care in 2010 [21]; mobile applications have been shown to improve adherence to referral for children with eye problems [22], enhance their compliance with wearing glasses to achieve the expected corrected vision [23], provide follow-up reminders as well as health education, and integrate professionals from different fields to supply health services [24]. Medication adherence was improved in glaucoma patients who used a smartphone medication reminder application [25]. However, a previous survey of mobile apps used in eye care indicated that mHealth mostly focused on eye diseases for elderly, and only 107 (22.53%) applications mentioned eye care professional involvement in application design or development [26]. Overall, the process of optimizing the clinical management of myopia would benefit from an alignment of best practice patterns with a tailored approach that can be achieved with close collaboration amongst eye care professionals (ECPs).

## 2. Methods

### 2.1. Study Goal

In this study, we aimed to develop and assess the usability of smartphone app-delivered OK lens care to increase compliance in parents with 7–12-year-old children wearing OK lenses and adherence to follow-up visits by providing individualized health education, feedback of tracking records, and reminder broadcasts through the app. Preliminary outcomes including compliance, ocular symptoms, and change in axial length were analyzed in this study. 

### 2.2. Study Design

This mixed methods study included needs assessment, design sessions, and feasibility and utility assessment. The institutional review board at Chang Gung Memorial Hospital approved all study procedures (no 201902199B0A3), and written consent was obtained from all study participants. 

Phase I: Needs Assessment

The research team reviewed the articles and guidelines of OK compliance and adopted a multifaceted approach to collect the management needs. The four steps of needs assessment included the following (Table 1): 1. interviewing 20 parents with children aged 6–13 years wearing OK lenses; 2. analyzing the social media posts of parents with children wearing OK lenses; 3. surveying 253 parents with myopic children on their compliance with wearing OK lenses; and 4. collecting the experience in managing OK treatment of ECPs by email (see Appendix A for interview guidelines). We outline the design of the four steps in Table 1. The detailed study protocol and findings of steps 1 and 2 were presented in previous studies [27,28]. We integrated these results and then developed a workflow of OK compliance management (Figure 1). Subsequently, after a draft paper-and-pencil prototype was established jointly by the principal investigator and the development team, feasibility and utility assessment was carried out.

Phase II: Design Sessions and App Development

For effective myopia control, children are required to wear OK lenses until they are teenagers or even older; thus, ensuring willingness for commitment from both parents and children, establishing a good rapport, and gaining the trust of patients/parents are very important. Based on this premise, the contents of the app were developed based on the results of the needs assessment from parents and were discussed in 2019–2020 with a multi-professional group comprising optometrists, ophthalmologists, nurses, and an information technology (IT) engineer. The app was named “EYE is OK” (since the English word “eye” and the Chinese character “love” have the same pronunciation), and the framework was based on the mHealth framework [21] including tracking records, individualized education, and reminders and required mutual engagement by parents (users), case managers (optometrists/caregivers), and ophthalmologists.

Wireframes were developed by an information technology (IT) engineer based on the prototype and the draft images of app characters and scenarios. We implemented the app build in a series of design sprints over a 5-month period. Once the user interfaces were determined, the research team then designed the administrator interface and the cloud computing algorithm. The administrator interface, accessed in webpages, included patients’ personal data and medical records, broadcasts, and individualized health educational data. Design sprints are time-constrained, phased periods in which specific programming work is completed, reviewed, and revised [29]. The final review was conducted by the entire advisory group of investigators in the last sprint with feedback and revisions. The end product of this phase was a beta version of the app ready for usability testing.

Phase III: Feasibility and Usability assessment

Usability testing of the app was conducted with 35 key stakeholders by conventional sampling of 30 parents, 2 optometrists, 2 ophthalmologists, and 1 nurse from three ophthalmology clinics. They received a download link of the software as well as a standard usage tutorial in video form in the invitation email. The purpose of this test was to discover any problems in the app and collect users’ feedback on various sections of the app as well as its management workflow. We adopted the System Usability Scale (SUS) to measure the usability consisting of 10 items [30]. Moreover, we also used a qualitative usability testing approach with 2 cycles of parents’ feedback and modifications. A research assistant introduced the app to the test participants who were asked to comment on words, to interpret phrases in the questionnaire, and to suggest improvements to the interface. A software engineer modified the app according to the feedback. The app prototype was modified until no further changes were suggested.

### 2.3. Participants

In phase III, an ophthalmology clinic in northern Taiwan was selected as the pilot site. The inclusion criteria for parents were the following: (1) being one of the parents who takes the responsibility of caring for the OK lenses of their child below 12 years of age with myopia (spherical equivalent (mydriatic) between −5.00 and −0.50 D (both eyes); (2) astigmatism (mydriatic) no greater than −1.50 D (both eyes); anisometropia of both eyes no greater than −1.50 D; (3) best-corrected visual acuity ≥0.00 log MAR units (Snellen equivalent to 20/20); (4) no diagnosis with any developmental diseases of the eye or the optic nerve system; (5) living in Taiwan with myopic children; (6) being able to read Chinese. Beside including the parents, five ECPs participating in phase I were also invited for feasibility and usability assessment. 

Parents along with their myopic children wearing OK lenses who visited the ophthalmology clinic were invited to install the beta version of the app to their own Android or IOS smartphones for usability testing. After 2 weeks, they received an email that reminded them to use the app and asked them whether they had experienced any technical problems. Subsequently, the 3-month usage data were collected for analysis, and parents were invited to complete the SUS survey during the first and last months of the trial.

### 2.4. Measurement

#### 2.4.1. Feasibility and Usability Assessment

1. System Usability Scale (SUS): We adopted the SUS to measure the usability consisting of 10 items [30]. Participants ranked each question from 1 to 5 based on how much they agree with the statement they are reading: 5 meant they agree completely; 1 meant they disagree vehemently. The participants’ scores for each question were converted to a new number, added together, and then multiplied by 2.5 to convert the original scores of 0–40 to 0–100. Higher scores reflect higher usability. An SUS score of at least 62.7 was considered acceptable, and 68 or above was regarded as above average in terms of usability quality [31]. The content validity index (CVI) and the internal consistency of the SUS were 0.95 and 0.90, respectively.

2. App satisfaction: We evaluated the app system satisfaction with the quality of the information and interface for parents and ECPs. All app functions and protocols were listed as items with scoring on a 5-point Likert scale from 1 (strongly satisfied) to 7 (strongly dissatisfied). High scores indicate high app satisfaction.

#### 2.4.2. Primary Outcome

1. Chinese Version of Orthokeratology Compliance for Myopic Children, OCMC: This measure of OK compliance behaviors had been used in a previous study conducted by the same research team [28] with acceptable reliability and validity. Moreover, the measurements of Jiang et al. [17] were integrated into the present study by including seven items for wear and care, and one item for compliance on the follow-up visit. The compliance rate was calculated by adding the number of days that each behavior is observed and then dividing it by the number of days the behavior should be performed according to the guidelines.

2. Ocular symptoms: Symptoms during OK lens wear at night, which were initially used by Yang et al. [9], were measured in this study. Six symptoms, i.e., inability to fall asleep, itchy eyes, redness, foreign body sensation, dryness, and pain, were included. Parents were required to fill in the number of days any of the above symptoms occurred per week. 

#### 2.4.3. Secondary Outcome

1. Axial length: The axial length of the eye was measured by a non-contact optical biometer (Tomey oa−2000) during the follow-up visit.

2. Follow-up visit: According to the clinic’s follow-up records, follow-up visits with an interval of less than fourteen days were categorized as regular follow-ups, those with an interval between fourteen days and three months were categorized as irregular follow-ups, and those with an interval of more than three months were categorized as no follow-up.

### 2.5. Statistical Analysis

Records of the compliance and the eye symptoms completed by the user were first downloaded from the app. Analyses were conducted using SPSS Statistics Version 22 (IBM Corp) for Windows (Microsoft Corp). The McNemar chi-square test and Wilcoxon signed rank test were used for comparison of outcome variables between baseline data and the 3-month assessment findings. Interview data were transferred into verbatim transcriptions and then categorized according to sentences and paragraphs. The results were then verified by the co-authors, and any inconsistencies were resolved by discussion and re-categorization if necessary.

## 3. Results

### 3.1. Development of the Compliance App

Based on the literature and OK practice guidelines [13], we developed the app framework containing four domains (Table 2), which were as follows: 1. clean and care ability; 2. ocular symptom observation and recording; 3. follow-up reminder; and 4. control indicator recording and monitoring. Furthermore, we integrated the results of the needs assessment to establish the app features and content (Table 2). The detailed results of steps 1 and 3 were presented in previous studies [27,28]. We illustrate the results of steps 2 and 4 here.

Step 2 employed thematic content analysis to investigate peer exchanging discourse conveyed in a closed social media (Facebook) group (“Ortho-K let me vision OK”) for children wearing OK lenses from October 2018 to March 2021. After excluding unrelated posts on children wearing OK lenses, 827 posts were analyzed by inductive thematic analysis to explore the comment threads to gain an in-depth understanding of considerations on children wearing OK lenses. The top-ranked themes categorized from posts were problems with assisting the wearing process for children (28.5%), non-compliance of intensive care (25.6%), and red or allergic eyes (16.9%). 

In step 4, we collected the experience and suggestions of providing care of OK lenses from ECPs. Beside the similar findings to steps 1 and 2, two ophthalmologists mentioned the importance of monitoring the axial length and the need for technological assistance to remind parents and their children to adhere to the follow-up visit. 

These results were subsequently implemented into the app, the detailed functions of which are shown below.

#### 3.1.1. End-User

The primary users of this app are the parents of myopic children, who are required to download the app, register, and complete the following steps sequentially.

1. Fill in personal and child’s basic information.

2. Compliance record: Record child’s wear and care status as well as the incidence of any eye symptoms prior to the follow-up visit. This record is entered once per follow-up. Parents can review their past compliance records as needed.

3. Record: Data visualization and analysis of several trends, including compliance, record of follow-up, and axial length control zone, are available to users.

4. Health education information: Individualized personal hygiene education information is provided based on the input information, including wear and compliance status, eye symptoms, and axial length changes. In addition, users can track the axial length information in the app at any time and adjust the child’s vision behavior accordingly.

#### 3.1.2. Managers

The administrator interface, built on a webpage, includes sections such as medical information, personal information, questions and answers, point redemption, broadcasts, charts, and health education information.

1. Medical information: This is where the administrator enters the axial data of the current visit as well as the date for the next visit.

2. Personal information: Only certain children’s and parents’ information is completed by the parent. Other information such as the date of lens replacement and the diagnosed degree of myopia is entered by the administrator.

3. Questions and answers: Any system or care questions raised by the user will be sent to and answered by the administrator via emails.

4. Incentive record: In order to improve app engagement, participants are rewarded with convenience store coupons with a value of TWD 50 whenever they complete surveys, read health education information, or attend a follow-up visit.

5. Broadcasts: This system includes pre-defined and manually pushed notifications. According to pre-defined time points, the system automatically transmits former notifications, including broadcasts prior to the follow-up visit, the release of new health education information, and health education messages unread for over a month. Manually pushed notifications are individualized health education information individualized for specific users.

6. Visual charts: Various data including basic information, wear and wash, care, comorbidities, and axial length records can be exported into Excel files in this section.

### 3.2. Developer and Cloud Computing

1. Reminder and alert: Reminders of follow-up visits are sent out automatically by the system a week before the visit, three days before the visit, and a week after an unattended visit.

2. Control computing: According to the literature, axial length growth is defined as the axial length of the current visit minus that of the last visit divided by the number of months between the two visits, which is then used to estimate annual axial length growth. Annual axial length growth (increase of <0.20 mm/year) [32] larger than the normal physiological annual growth is categorized as the “red zone”. Alternatively, annual axial length growth larger than zero but smaller than the normal physiological annual growth is categorized as the “yellow zone”. Last, axial length that remains even or is even reduced is categorized as the “green zone”. In addition, patients are divided into “Poor compliance”, “Moderate compliance”, and “Complete compliance” groups and categorized into the “red zone” (0–50%), “yellow zone” (51–99%), and “green zone” (100%), respectively, according to their degree of compliance with the medical instructions.

### 3.3. End-User Testing, Usability, and Feedback

A total of three months of trial data were collected from 30 parents and their children. The results show that most users were mothers (N = 27, 90%) and had an education degree of college or higher. Of these parents, 93.3% had myopia, and 14.2% had high myopia (SE ≤ −5.00 D). Alternatively, the demographic data of children wearing OK lenses indicated that most children were diagnosed with myopia when they were below 10 years of age (N = 21, 70%). In addition, the mean age of myopic children was 11.0 years (SD = 1.33), and most children had been wearing OK lenses for a period of 6–12 months (N = 16, 53.3%). The most common degree of myopia was –3.00~4.00 D (N = 18, 60.0%) (Table 3 and Table 4).

#### 3.3.1. System Usability Scale (SUS) Scores

To provide context for the SUS score, the adjective rating scale proposed by Bangor et al. [31] was used, where a score of 39.17 to 52.00 is considered “poor”, 52.01 to 72.74 is considered “OK”, 72.75 to 85.57 is considered “good”, and 85.58 to 100.00 is considered “excellent”. As shown in Table 5, the SUS scores of the 30 parents and the 5 ECPs were 73.4 (SD = 9.3) and 73.5 (SD = 11.2), respectively, falling into the “good” category. More specifically, ECPs scored poorly in “I’d imagine most people will quickly learn how to use the system” out of all the questions. In contrast, parents scored uniformly across different questions, with an average score of over 3.8 in all questions. After three months of app engagement, the SUS scores of the 30 parents and the 5 ECPs were 82.5 (SD = 7.2) and 83.6 (SD = 6.3), respectively, both showing significant improvements. Parents and ECPs clearly identified the features, benefits, and risks of an app for compliance with OK lenses and incorporation into clinical care.

#### 3.3.2. Qualitative Feedback and Satisfaction with the App

Qualitative feedback on three domains with 16 functions was collected from ECPs, and only feedback on the user interface of the app was collected from parents. The results show that the features of the reminder and axial length record gained the highest satisfaction scores by parents and ECPs. ECPs were highly satisfied with the date for the next follow-up visit of the administrator interface and the cloud computing for axial length change. Qualitative feedback on the app is listed in Table 6 below. All suggestions provided by the parents and ECPs were modified through testing to form the final version.

### 3.4. Preliminary Outcomes

Compared with the baseline, three months after app engagement, the average number of days the 16 children performed wear and wash per week increased from 6.12 days to 7 days as recorded by the parents; the percentage of care completion increased from 88% to 92%. Itchy eyes, redness, and foreign body sensation were the three top symptoms observed by parents, and there were no significant changes in the 3-month testing (Table 7). Axial length changed from 24.21 to 24.25 mm (with an average increase of 0.03 mm), and the 3-month follow-up rate was 100% (Table 6).

## 4. Discussion

### Principal Findings

To our knowledge, the app developed in the study is the first app for long-term management of myopia treatment in children. Mobile-based management that permits real-time data collection, education, and sharing is becoming increasingly commonplace, enabling ECPs to monitor axial length growth for myopia control and to provide useful suggestions for parents. The “EYE is OK” app is the first smartphone app with interactive online management for myopia control. Although many of the small-scale studies published thus far provide promising evidence supporting the potential of mobile apps to positively impact adherence and treatment outcomes among children and young people [20], most existing ophthalmology-related apps are limited to diagnosis, screening [24] or reminding [33], and data collection for dry eye [34]. Our app includes functions such as recording, reminding, and education, and it requires trained ECPs to review the patient’s compliance behavior before a follow-up, record the axial length during the current visit as well as the date for the next visit, and provide the patient with tailored health education information, thereby improving the patient’s compliance behavior, reducing the risk of comorbidities, and minimizing axial length growth. In addition, the management process and the content of the app were established based on needs assessment of OK care that was collected both subjectively and objectively, allowing an interactive management model between users and administrators. Although users’ feedback on the app was good, qualitative feedback from users and administrators indicated that there is still room for improvement. More specifically, whether the axial length and the date of the next follow-up visit should be entered by the user or the administrator remained controversial. Due to the high prevalence of myopia in Taiwan, ophthalmology clinics have been overloaded with the diagnosis referred by school screening and treatment of myopic children. Even when the rate of OK lens wear is 10%, it remains a heavy burden on the human effort of these clinics. As innovative management models, smartphone apps are a global, convenient, cheap, and interactive strategy for myopia management. In addition, apps of individualized professional management have been proven more effective in changing the patient’s behavior than those relying on the user’s independent management [25]. The success of an app also relies on the balance between management effectiveness and user convenience. Therefore, the app developed in this study can be further improved to help users achieve their purposes and avoid being one of the many apps that have an extremely low download rate. Furthermore, self-managing apps cannot achieve the desired result among patients with poor self-control, in which case an interactive model with professionals is required [26]. Therefore, another way to improve the current app is to develop multiple versions. For example, a parental version and a professional version can be developed, of which the former would be managed by parents, whereas the latter would be managed by professionals, thereby distinguishing between different users.

The compliance records were self-reported by participants. In this study, this method was adopted to record children’s care and wear behaviors. Although, theoretically, this record should be entered daily, previous research often collected data on a weekly basis. In this study, the frequency of data collection was initially set monthly but was later changed to a certain week during the month before the follow-up, as users complained about too many items and frequent entries. Through broadcast reminders, the rate of regular follow-up was 80%. Correct completion of compliance records is compulsory for individualization of health education to achieve care goals. Previous research has also found that non-compliance behaviors vary with different wearing times [15,28]. In addition, the OK lens wearing process is the most frequently asked question by parents in the Facebook community. Therefore, a total of 25 items were initially included in the compliance behaviors in this study. The number of items was first changed to 27 according to experts’ recommendations but was then reduced to 24 in the final version, as parents complained that there were too many items. This reveals an issue of the interactive design mode, i.e., that both tracking effectiveness and user convenience must be taken into consideration when designing the app.

## 5. Conclusions

In conclusion, the “EYE is OK” app is an important platform that helps manage the use of OK lenses in myopic school children. It not only improves parents’ ability to assist children in wearing OK lenses but also promotes regular follow-up visits. In addition, by using cloud computing algorithms to divide patients into different risk categories according to their axial length growth, the app can provide users with tailored health education and consequently help them comply with medical instructions. Through such designs, the app received positive feedback from both the users and the ECPs. However, as this app is an interactive management app that requires a dedicated ECP as the app administrator, a key component of its success is appropriate human effort deployment in the ophthalmology clinic.

## 6. Limitations and Future Directions

The EYE is OK app plays an important role in helping parents and ECPs manage the compliance with OK lenses for parents with myopic children. In addition to the improvements proposed by the users, there are limitations of this study that should be overcome in future research. For example, this study was conducted in one clinical setting, making it infeasible to understand the applicability of the app in different settings. Therefore, recruiting participants from multiple centers should be included in future research to enhance the usability of the app. Furthermore, in the future, the app should be separated into a parental version and a professional management version. In addition, the sample size of random trial research can be adopted to demonstrate the effectiveness of different versions of the app.

## Figures and Tables

**Figure 1 ijerph-18-10316-f001:**
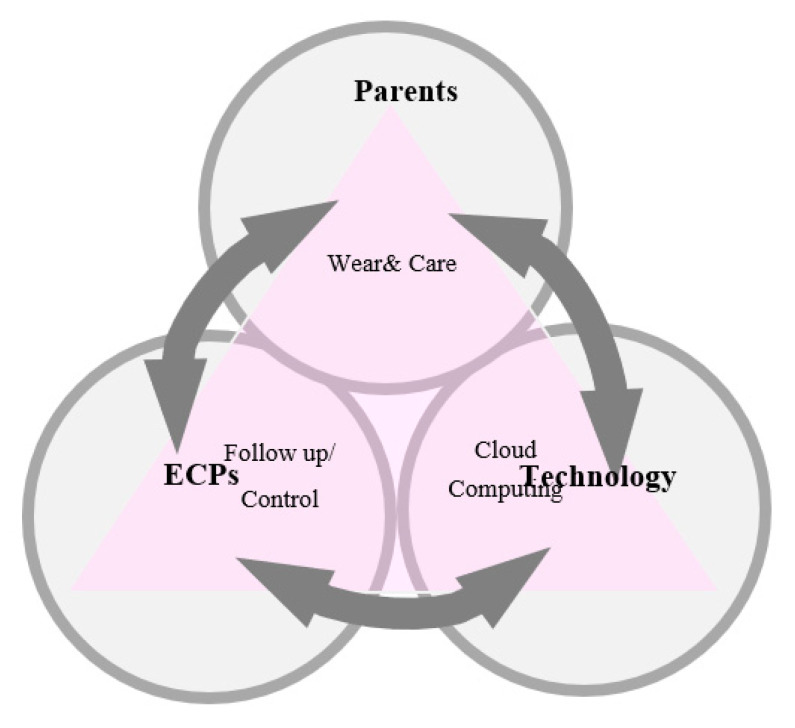
The workflow of orthokeratology compliance management. Note: ECPs—eye care professionals.

**Table 1 ijerph-18-10316-t001:** The detailed study design for the four steps in the needs assessment.

Steps	Step 1	Step 2	Step 3	Step 4
Time	2019.8–2019.12	2020.3–2020.6	2020.3–2020.7	2021.8–2021.12
Method	Qualitative	Qualitative	Quantitative	Qualitative
Sampling	Parent who is mainly responsible for complying to OK lens care for their myopic child aged 6–13 years.	A closed social media (Facebook) group (“Ortho-K let me vision OK”)	One of the parents who takes the responsibility of caring for OK lenses of their child with myopia below 12 years old.	Eye care practitioners who have experience in caring or prescribing OK for myopic children for at least three years.
Purpose	To illustrate the decision-making process, the experience of wear and care for myopic children and the compliance with OK lenses.	To collect the concerns, questions, and care experience of OK lenses for myopic children.	To analyze the compliance of parents with OK care for their young children.	To collect professionals’ opinions of OK compliance of myopic children aged 7–12 years.

Note: OK—orthokeratology.

**Table 2 ijerph-18-10316-t002:** Needs assessment results and app functions.

Steps	1	2	3	4
Participants	16 mothers/3 fathers	827 posts from 2018 to 2021 had been analyzed	253 parents	2 optometrists, 2 ophthalmologists, and 1 nurse
Main themes				
Wear and Care	Unfamiliar with the steps Worried about breaking the lenses Losing the lenses	Problems with assisting wearing process for children (28.5%) Choosing solution (14.4%)	68.6% for wear and care behaviors	Worried about breaking the lenses Losing the lenses
Intensive Care	Forgetting or ignoring the steps	25.6% of posts discussed non-compliance with intensive care	25.0% adhere to the procedure	Ignoring the steps
Ocular Symptom	Red or allergic eyes	Red or allergic eyes (16.9%)	60.5% of parents observed ocular symptoms	Parents need to be aware of eye symptoms of their children
Control Indicator	Unaware of control effect Can see clearly VA is normal	Caring about VA and clear vision in the daytime (12.6%)	18.2% did not know that the axial length changes as an effect of myopia control	Parents and ECPs need to acknowledge the importance of monitoring AL length
Follow-Up Visit	Regular wearing in the first year Reminder needed		Follow-up visits significantly increased with provision of axial length information and knowledge of axial length changes as an effect of myopia control	Technological assistance to remind parents and their children to adhere to the follow-up visit
Application Design	Health education contents Management process	Health education contents	Reminder mechanism Feedback on medical records Management process	Health education contents Management process

Note: VA—visual acuity; AL—axial length; ECPs—eye care professionals.

**Table 3 ijerph-18-10316-t003:** Demographic information of parents.

Variable	N	%
Parents		
Fathers	3	10.0
Mothers	27	90.0
Educational level		
College	4	13.3
University and above	26	86.7
Myopia		
No	2	6.7
Yes	28	93.3
Spherical equivalent refraction (SE) of right eye (diopters, D) (range, −0.50 to −16.00 D)		
≤−5.00	4	14.2
>−5.00	24	85.8

**Table 4 ijerph-18-10316-t004:** Demographic information of myopic children.

Variable	N	%
Child age (range, 6 to 13 years)	Mean = 11.1	SD = 1.33
<10 years old	3	10.0
≥10 years old	27	90.0
Onset age (range, 6 to 12 years)	Mean = 8.35	SD = 1.50
<10 years old	21	70
≥10 years old	9	30
Spherical equivalent refraction (SER) of right eye (diopters, D) (range, 0 to −7.50 D)	Mean = −2.39	SD = −1.32
>−2.00D	1	3.3
−2.00 to −3.00	9	30.0
−3.00 to −4.00	18	60.0
−4.00 to −5.00	2	6.7
Period of wear (months)	Median = 9	
≥1~6	6	20.0
≥6~12	16	53.3
≥12	8	26.7

**Table 5 ijerph-18-10316-t005:** System Usability Scale (SUS) scores.

Domain	ECPs		Parents	
	Mean	SD	Mean	SD
Total SUS score				
First month	73.5	11.2	73.4	9.3
Third month	83.6	6.3	82.5	7.2
Wilcoxon signed rank test	3.84 *		3.84 *	

Note: ECPs—eye care professionals; * *p* < 0.05 (two-tailed).

**Table 6 ijerph-18-10316-t006:** Feedback of features and functions of the EYE is OK app.

Domain	Functions/Features	Suggestions	Satisfaction
User Interface	Login/Register		
	1. Basic information input	E: Fill in the time for next lens replacement P: Cannot remember the degree of myopia	Mean = 3.98 (P) Mean = 4.00 (E)
	2. Reminder	P: The icon is too small and unclear	Mean = 4.10 (P) Mean = 4.80 (E)
	3. Compliance—care and wear	E: Should indicate the solution required for each step P: Too many items in the questionnaire	Mean = 3.76 (P) Mean = 4.00 (E)
	4. Compliance—intensive care	P: Some tasks are completed by the child	Mean = 3.76 (P) Mean = 4.00 (E)
	5. Ocular symptom record	P: Need to record the date	Mean = 4.00 (P) Mean = 4.00 (E)
	6. Axial length record	E: Add multi-period query	Mean = 4.27 (P) Mean = 4.8 (E)
	7. Health Education	Nil	Mean = 4.00 (P) Mean = 4.27 (E)
	8. Questions and Answers	Nil	Mean = 3.76 (P) Mean = 4.00 (E)
	9. Incentive system	Nil	Mean = 3.90 (P) Mean = 3.80 (E)
Administrator Interface	10. Date for the next follow-up visit		Mean = 4.8 (E)
	11. Axial length recorded in the current visit	Nil	Mean = 3.80 (E)
	12. Health education information	Nil	Mean = 4.00 (E)
	13. Broadcasts	E: Add individualized notifications	Mean = 4.00 (E)
Cloud Computing	14. Axial length change	P: Serves as a reminder	Mean = 4.8 (E)
	15. Degree of compliance behaviors	P: Not very helpful	Mean = 4.0 (E)
	16. Broadcasts—follow-ups/health education	Nil	Mean = 4.0 (E)

Note: P—parents; E—eye care professionals.

**Table 7 ijerph-18-10316-t007:** McNemar test and Wilcoxon signed rank test of outcome variables.

Variable	Baseline	3-Month	χ^2^/z
Full compliance	82%	85%	1.62
Wear and wash	90%	92%	1.17
Intensive care	72%	88%	1.92
Regular follow-up visits	78%	100%	4.25 *
Ocular symptom—itchy eyes	3	2	0.08
Ocular symptom—redness	2	1	0.06
Ocular symptom—foreign body sensation	2	1	0.06
Axial length growth	24.21	24.25	0.76

* *p* < 0.05.

## Data Availability

The data presented in this study are available on request from the corresponding author.

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
