# Peer review of "A Cooperative Management App for Parents with Myopic Children Wearing Orthokeratology Lenses: Mixed Methods Pilot Study"

_ijerph, 2021, doi:10.3390/ijerph181910316_

Round 1
Reviewer 1 Report
The authors developed a mobile application to improve compliance in orthokeratology. There were detailed descriptions about the background, study design and development stages. However, the manuscript is more like a report on the strategy in the development of an application, rather than a scientific paper reporting the qualitative/quantitative findings as the results were not well supported by evidence (data).
Part I was about needs assessment. It was a qualitative study about the results of needs assessment from group interviews. The authors just listed the items identified in Table 1 without presenting the flow of how these items were drawn without any data (e.g. response rate) to support the findings.
Part III was about the feasibility and utility assessment. The beta version of the application was tested in 30 parents and 5 ECPs. The authors did not give any details on the methods used to collect feedbacks, e.g. use of questionnaires. If questionnaires were used, then what questions have been asked? The SUS scale was used but the questions asked were not presented. Thus, results presented in Tables 4 and 6 just jumped out from results. Any the feedback shown in Table 5 did not have any response rate to support the problem. For example, there were only 5 ECPs involved in the test. Concerns identified in Table 5 were from only one or from more than one ECPs? About the small sample size, paired t tests employed in Table 4 may not be appropriate.
Another example about lacking of data to support claim is as follows:
In the abstract, the authors said that ‘parents were “most” satisfied with the app’s reminder….’. However, the current results only identified outcomes. The claim ‘most satisfied’ was not supported as no rating was obtained.
Others comments.
Spaces were missing between words (e.g. line 2 ‘WithMyopic’; line 38 ‘leadsto’ etc.) or before references (e.g. line 73 ‘behaviors[19]). This problem occurs in the whole pdf file which is annoying for reading.
Line 50. Define ‘OK’ for the first appearance in the manuscript
Line 64. Regular OK aftercare ensure good refractive correction and ocular health, not necessarily the growth of eyeball. Only true if this is routine in OK aftercare which may not be applicable in many myopia control practice
Line 71. Incomplete sentence
Line 88. Define ‘ECDs’ for the first appearance in the manuscript
Line 171. Replace ‘Ortho-K’ with ‘OK’
Line 228. Subheading for ‘End user’?
Line 279. Subheading for ‘End-user testing, usability and feedback’?
Line 383. Replace EYE is OK with ‘EYE is OK’
Figure 2. Consider to remove as it’s not relevant and important
Figure 3. Resolution too low
Table 3. (1) two decimal places for refractive error (2) replaced 'diopters' with 'diopters, D'
Table 4. Spell out SUS in the table title and define ECPs in the table
Reviewer 2 Report
Summary
The article describes how to develop an app that is helpful for parents and children who wear OK lenses for myopia control. When wearing OK, it is important to follow certain hygiene rules and regularly assess the integrity of the eye and the lens-eye alignment. To strengthen the awareness of such basic aftercare concepts adherence to the rules, the developed app should help to increase the compliance.
In previous studies or in clinical practice, patients often use a simple contact lens-diary so that the practitioner get an idea of the wearing schedule. The idea of developing an app for this is good. Storing additional health data such as eye length in the app is very useful for myopia control. This makes it possible to better monitor the compliance and the success of the therapy and to read out a correlation between compliance and the success of the therapy over a longer period.
In the method section information is given which steps were done to design and develope this app.
The results chapter should be rewritten and better structured. The authors should think carefully about the results they want to show. For example, the results of how the app was designed and in what form, or the results of the clinical trial and the evaluation of the information that the parents entered into the app.
In the current version, this chapter is difficult to read and is not logically structured.
Table 1, for example, is very difficult to read. These individual sentences do not make a claim. It should be stated how often these statements were made so that the reader can get an idea of how the app was rated by the test participants.
Furthermore, the evaluation of the oculer symptoms is missing, for example. According to line 194 in the methodology, this information was requested and entered into the app. However, no results can be found.
Line 51: OK is not the only optical method to slow the progression of myopia. Additionally, you should clarify what you mean with “correct refractive errors”. Soft contact lenses also correct the refractive error and give clear vision during the day with the lens in-situ. However, the benefit of OK is that during the day vision is corrected without a contact lens (OK) in place.
Please review the literature an make it more precise what you mean.
Line 71: Poor without a capital letter.
Line 127: Space missing between parents and
Line 129: Space missing between important. Based
Line 271: What is according to you a normal physiological annual axial length growth? Please refer to a reference.
Line 282: Please define what is high myopia?
Round 2
Reviewer 2 Report
Dear authors
Thank you for the revision and improvement of the manuscript. Adding and improving the tables in the results chapter makes the study more readable and logical.
I think the paper can be published in this way.